# Bioelectrical Impedance Analysis of Oral Cavity Mucosa in Patients with Lichen Planus and Healthy Controls

**DOI:** 10.3390/dj10070137

**Published:** 2022-07-19

**Authors:** Christian Bacci, Alessia Cerrato, Anna Chiara Frigo, Matteo Cocco, Gastone Zanette

**Affiliations:** 1Oral Medicine and Pathology Unit, Clinical Dentistry Section, Department of Neurosciences, University of Padova, 35128 Padova, Italy; cerrato.alessia92@gmail.com (A.C.); matteo.cocco93@libero.it (M.C.); 2Departments of Cardiac, Thoracic, and Vascular Sciences, and Radiology, University of Padova, 35128 Padova, Italy; annachiara.frigo@unipd.it; 3Dental Anesthesia, Clinical Dentistry Section, Department of Neurosciences, University of Padova, 35128 Padova, Italy; gastone.zanette@unipd.it

**Keywords:** cancer, diagnostic procedure, oral lichen planus, oral cancer, oral lesions, oral pathology

## Abstract

Objectives: Oral lichen planus (OLP) is an inflammatory disease. Bioelectrical impedance analysis (BIA) is a method for assessing tissue composition. Based on a combination of reactance and resistance data, a phase angle is calculated that may range from 90° to 0°, and that correlates with body cell mass. There is evidence to suggest that neoplastic tissue has a lower phase angle than normal tissue. The aim of the present experimental study was to establish whether OLP patients have a different tissue phase angle from healthy controls. Materials and Methods: Bioelectrical impedance measurements were obtained for the buccal mucosa, tongue, hard palate and upper anterior gums using an ad hoc device in a sample of 57 consecutive patients with OLP and 60 healthy controls, and their phase angles were calculated. Results: The mean resistance, reactance, and phase angle of the hard palate and gums were higher in the OLP group than in the controls, and the differences were statically significant. The resistance and reactance recorded for the adherent gingiva and hard palate were always higher in the OLP group (*p* = 0.044; *p* = 0.020; *p* = 0.054), and so was the phase angle for the adherent gingiva. No statistically significant differences emerged for the other areas of the oral cavity (*p* < 0.05). Conclusion: These findings confirm differences between the bioelectrical impedance of OLP lesions and that of healthy oral tissues. Clinical relevance: Bioelectrical impedance analysis could be useful in the diagnosis of OLP.

## 1. Introduction

Oral lichen planus (OLP) is a chronic inflammatory disease that affects the oral mucous membranes. It has a variety of clinical presentations, and its pathogenesis is still unclear [1,2].

The most common sites of OLP are the cheeks, tongue, and adherent gingiva. The palate, oral floor and labial mucosa are less frequently involved [3]. OLP lesions are generally bilateral, while single sites of OLP are uncommon [4]. The clinical manifestations of OLP vary, and Andreasen classified six different types: reticular, papular, plaque, atrophic, erosive, and bullous [5]. A simplified classification proposed by Eisen distinguishes only between reticular, erythematous and erosive OLP lesions [6]. The incidence of carcinoma associated with OLP is reportedly less than 0.5%, although some studies have indicated percentages of neoplastic transformation ranging from 0% to 5.3%. Some authors have suggested that the diagnosis necessitates histological examination, and this has become mandatory to comply with the AAOMS criteria [7,8,9,10,11,12].

Bioelectrical impedance analysis (BIA) of body composition is used to assess lean mass, fat mass, and fluids in a biological conductor. BIA involves applying a harmless low-intensity (400–800 μA) alternating electrical current to organic tissues and measuring the intensity of the output current. The characteristics of a body’s electrical conductivity can influence the output current being measured. The first step in BIA involves recording the electrical resistance and reactance of a whole body or body segment. These data are acquired by attaching four electrodes: two are “injection” electrodes used to deliver an electrical current of known, constant intensity and high frequency through the body; the other two are “sensor” electrodes that identify the difference in electrical potential induced by the current’s passage through the tissues. From the physical standpoint, bioelectrical impedance is the sum of two components, i.e., resistance and reactance, according to the formula [13]:Z^2^ = R^2^ + Xc^2^
where Z is the impedance, R is the resistance, and Xc is the reactance.

The BIA 101 ANNIVERSARY (AKERN, Montacchiello, Pisa, Italy) is an impedance vector analyzer for assessing a body’s hydration and nutritional status. In technical terms, the electrical properties of the tissues (resistance, reactance and phase angle) are detected in real-time, and the bioelectrical impedance of the different parts of the body is estimated. The classical BIA conducted with the 101 ANNIVERSARY analyzer employs the tetrapolar technique, with a sinusoidal current at a frequency of 50 kHz. The current is kept constant at 330 µA RMS on loads from 1 to 5000 Ω. To saturate the whole body with the current, the injectors are positioned distally on the metacarpal and metatarsal line using surface electrodes, while special four-pointed electrodes detect the segmental resistance, reactance and phase angle data inside the oral cavity, acting simultaneously as the points of entry and exit of the electric current so as to close the circuit.

The tips of the electrodes have a semicircular design to enable impedance measurements in the oral cavity. The “fork” is made of fiberglass, and its drop-shaped tips are made of tin. The prongs of the fork are 2 cm long. Two electrodes serve as injectors, and two as detectors (Figure 1 and Figure 2).

The aim of the present experimental study was to ascertain any differences in the bioelectrical impedance of tissues in OLP patients as compared with healthy controls.

## 2. Definition of Resistance, Reactance and Phase Angle

All substances oppose a degree of resistance to the passage of an electrical current. The electrical resistance of the human body is inversely proportional to the total amount of water it contains. This means that a well-hydrated individual will have a lower electrical resistance than one who is dehydrated [14,15]. Tissues containing little fat are good electrical conductors, offering a low resistance because they contain proportionally more water and electrolytes. Adipose tissues and bone are poor conductors because they are low in fluid content and offer high resistance to the passage of an electrical current. The electrical resistance of a tissue thus depends not only on the water content of the body as a whole but also on its proportional distribution in the various types of tissue [12].

Reactance is a force that opposes the passage of an electrical current to a degree that depends on the capacitance of a system (its capacity to store an electrical charge). In the human body, the cell membrane forms a thin (insulating) lipid layer that separates two environments, both rich in fluids and proteins (conductors). It is an excellent example of a capacitor, which is why the cell has a reactance that opposes the passage of an electrical current [11]. In practical terms, this phenomenon is common to all types of cells except for adipocytes, which are full of fatty material and therefore have good insulating qualities but lack capacitance. A body’s reactance is consequently proportional to the cellular mass of its lean tissues. It is important to add that cells must be in good physiological condition to develop a reactance. Cell suffering is generally associated with a gradual drop in the electrical potential of the membrane because the cellular pumps become less able to maintain an adequate gradient between the ions inside and outside the cell [11,13].

The phase angle is a linear value correlating the resistance with the reactance of an electrical circuit. The phase angle can vary between 0° and 90°. Lower phase angles are associated with low resistance or low reactance, which may mean an expansion of the extracellular spaces due to fluid retention, or a loss of cell membrane function due to necrosis, for example. Higher phase angles are associated with high resistance or high reactance and may indicate states of dehydration, or considerable quantities of intact cell membranes, for instance. The phase angle can thus serve as a prognostic indicator of cell membrane integrity. The use of BIA has enabled rapid advances in numerous fields, including dietology, nephrology [16], surgery, cardiology, hemodynamics, and sports medicine [17].

The present experimental study involved the use of a medical device specifically designed to measure the bioelectrical impedance of soft tissues in the oral cavity. A substantial difference in electrical impedance has been demonstrated between healthy and tumorous tissues in the oral cavity and between clinically healthy areas of the oral cavity and other areas showing signs of erosive OLP (lesions that can mimic tumors [18]). Tumors and areas of erosive OLP both coincide with a much lower phase angle than that of healthy tissue [19].

## 3. Materials and Methods

### 3.1. Study Population

Sixty-four consecutive patients with a clinically- and histologically-confirmed diagnosis of OLP according to the AAOMS criteria [5] were recruited to join our group of “cases”. An in clear investigation case-control pilot study was performed. Our null hypothesis was no difference in resistance, reactance and phase angle between healthy controls and OLP patients and between areas of oral mucosa with and without clinical evidence of lesions in OLP patients. The numerosity of the sample required was established a priori.

The patients enrolled in the study were being followed up at the Oral Medicine and Pathology outpatient service at the University of Padua (Italy), where BIA was conducted as part of routine clinical practice. Exclusion criteria for this group of patients concerned the use of non-removable electronic devices (pacemakers, internal defibrillators, or cochlear implants). A control group of individuals with clinically healthy oral cavity mucosa was enrolled, matched for age and sex, and with no medical history of diseases that might affect the oral cavity. All candidates for the control group also met the above exclusion criteria regarding non-removable electronic devices, and they underwent a clinical examination to establish the absence of any oral cavity lesions or signs of disease.

In the graphs and tables, any lesions are identified by means of numerical values: 0 (no lesions), 1 (striations), 2 (erythema or erosion) [5]. The areas of the oral cavity taken into consideration were: the right and left cheeks, the tongue, the hard palate, and the adherent gingiva. Lesions mapped during the study as erythematous/erosive with striations were included in the group of purely erythematous/erosive lesions because such lesions are still classed as erythematous/erosive in the literature, even in the presence of striations [20] or superinfections [21].

The bioelectrical impedance measurements were compared between the cases and controls. The temperature of the area of the oral cavity analyzed was recorded by the device, but it was disregarded for the purposes of our study because it is strongly influenced by room temperature as well as by that of an individual’s body (as explained in the device manufacturer’s instructions).

### 3.2. Analytical Method

BIA was conducted using a device with a special sensor, and four electrodes were placed over an area of the oral mucosa to record the bioelectrical impedance of the underlying tissue. All measurements were taken by a single operator (M.C.) who had been trained to use the device. The complete measurement procedure took a mean of 5 min. None of the patients or controls reported any discomfort caused by the procedure, which proved to be technically straightforward, quick to complete, and well tolerated. The measurement procedure involves resting the sensor on the area of the oral cavity of interest and waiting until the values it records become stable. These values are then written on a chart designed for said purpose by an assistant, who also recorded any signs of lesions characteristic of OLP. Measurements were taken systematically in five areas of the oral cavity: both cheeks, the hard palate, the tongue, and the adherent gingiva.

### 3.3. Statistical Analysis

The data collected were input into an EXCEL spreadsheet and analyzed using SAS 9.4 (SAS Institute, Cary, NC, USA) for Windows. Qualitative data were represented as counts and percentages in each category, and quantitative data were expressed as means and standard deviations, medians, and minimum and maximum values. Comparisons between groups and by the severity of the lesions (for OLP cases) were drawn using the Chi-squared test for qualitative variables and Wilcoxon’s rank-sum test or the Kruskal–Wallis test for quantitative variables (given their non-normal distribution). The Kruskal–Wallis test was followed up with the Dwass–Steel–Critchlow–Fligner method for multiple comparisons in cases of statistical significance. The level of significance was set at 5%, and the numerosity of the sample required was established a priori.

## 4. Results

There were 127 individuals recruited for the purposes of this study, 64 cases and 63 controls. A total of 7 individuals (4 cases and 3 controls) were excluded from the analysis because it proved impossible to obtain measurements of one or more areas of their oral cavities. It generally proved most difficult to measure the hard palate and adherent gingiva. After their enrolment, 3 individuals in the OLP group refused to take part in the study, citing various reasons (lack of time, little interest in the study proposal, and a history of hypersensitivity to numerous materials), so they only completed the part of the procedure needed for their own treatment. The two groups forming the object of the statistical analysis thus included 60 clinically healthy controls and 57 patients with clinically and histologically confirmed OLP. The group of OLP patients was a mean of 60.3 years old (minimum 23, maximum 86 years), and the control group was a mean of 63.1 years old (minimum 23, maximum 89). The OLP group consisted of 15 males (26.32%) and 42 females (73.68%), and the control group included 29 males (48.3%) and 31 females (51.67%).

In the study sample as a whole (cases and controls), BIA was used to measure 613 areas of the oral cavity, and 2772 numerical values were recorded on the charts. The data were input in two Excel spreadsheets, with one patient per line, and the various measurements were distributed in subsequent columns.

### Lesions Identified in OLP Patients

Measurements of the hard palate revealed no lesions in 53 patients, while 2 had striations, and 2 had erythematous lesions. Measurements of the adherent gingiva identified no lesions in 39 patients, while 5 had striations, 11 had erythematous lesions, 1 had plaque lichen planus, and 1 had erythematous lesions with striations. No lesions were identified in the left cheek of 21 patients, while 25 showed striations, 4 had pure erythematous/erosive lesions, 6 had erosive lesions associated with striations, and 1 had plaque. In the right cheek, there were no lesions in 19 patients; 28 had striations, 9 had pure erythematous/erosive lesions, and 1 had plaque lichen planus. As for the tongue, 43 patients revealed no lesions, 12 had striations (which have a very particular, characteristic appearance on the tongue, often showing atrophy of the papillae, striated areas, and white lesions), and 2 patients had erythematous lesions.

Table 1 summarizes the results of the analysis of the two groups as a whole, which are discussed below. There was a statistically significant difference between the resistance and reactance of the adherent gingiva and hard palate, with both values being higher in the OLP group.

On statistical analysis, there were significant differences in the resistance and reactance recorded for the adherent gingiva and hard palate, which were always higher in the OLP group (*p* = 0.044; *p* = 0.020; *p* = 0.054). There was also an increase in the phase angle in the case of the adherent gingiva. No statistically significant differences emerged for the other areas of the oral cavity (Table 2, Table 3, Table 4 and Table 5).

## 5. Discussion

As mentioned earlier, BIA has proved useful in various fields, prompting the present authors’ interest in applying the concept to the sphere of oral diseases. In patients with OLP, not all areas of tissue without clinically apparent lesions can be considered “healthy” in the sense of lacking any histological changes [22]. Patients with OLP may only have clinically appreciable lesions in some parts of the oral cavity, and the sites involved may vary in appearance over time [23]. Another characteristic of OLP lies in its potential for neoplastic degeneration. As reported in several published studies, this can occur in 0.4% to 1% of cases, so patients with OLP require routine follow-up. That said, the polymorphous nature of OLP may sometimes mean that any neoplastic transformation is mistaken for just another clinical appearance of the disease itself. For the time being, biopsy is the gold standard for differential diagnosis in this setting.

The variety of clinical presentations, male-to-female ratio (2 to 3), and mean age of the patients involved in this study are consistent with the epidemiological data in the literature. When healthy controls and patients diagnosed with OLP were compared, the phase angles were generally higher in the latter (see Table 1). The difference was only statistically significant when the resistance and reactance of the hard palate and adherent gingiva were considered. The phase angles for the cheeks and tongue were higher as well, but not to a statistically significant degree. This comparison did not distinguish between striations and erosive OLP lesions, however. Ultimately, the phase angles were higher in OLP patients than in healthy controls for all areas of the oral cavity except the tongue, for which the values in the two groups almost overlapped. Patients with higher phase angles might warrant closer follow-up so that any onset of OLP could be diagnosed early on. It might be very useful to test this issue in larger samples of dental patients in an effort to identify ranges of values capable of pinpointing cases of OLP, or at risk of OLP, and consequently warranting routine follow-up. These results confirm the findings of a previously-published study [19].

Measuring bioelectrical impedance could also be useful in the follow-up of erythematous/erosive lesions for the purpose of identifying ranges of bioimpedance values associated with tissues that are already dysplastic or neoplastic on histological examination.

A significant difference in the temperature of the oral mucosa emerged between our healthy controls and cases of OLP: it was lower in the latter, as also reported in other published studies. After consulting the manufacturer of the device used for BIA in our study, however, this indicator was judged to be of little value because it is strongly influenced by room temperature.

## 6. Conclusions

Given the adequate numerosity of the lesions considered in our sample of OLP patients, the present study demonstrated that BIA can identify anatomical and physiological changes in the mucosa of the oral cavity.

## Figures and Tables

**Figure 1 dentistry-10-00137-f001:**
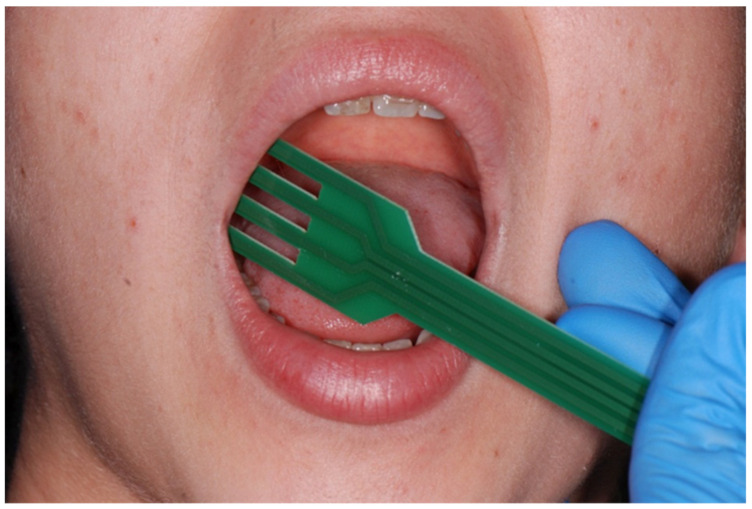
The use of the “fork”.

**Figure 2 dentistry-10-00137-f002:**
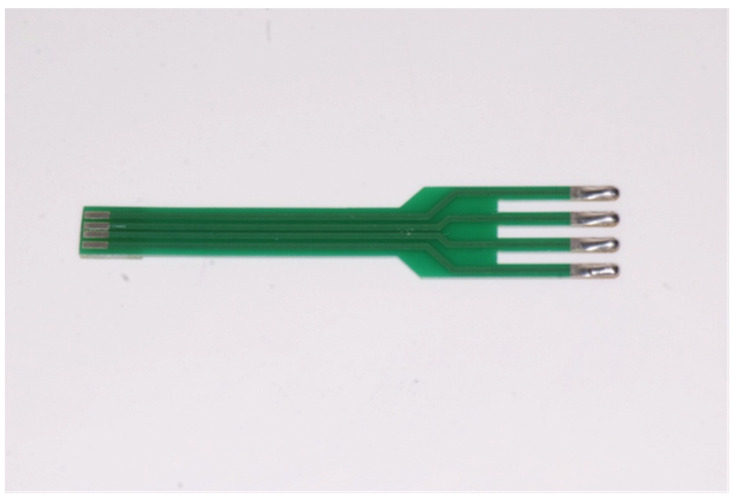
“The fork”.

**Table 1 dentistry-10-00137-t001:** Comparison between measurements in the OLP group and healthy controls (means and standard deviations). Electrical values are in Ohm. 1 = Wilcoxon’s rank-sum test.

Variable	*p* Value	Control Group (*n* = 60)	OLP Patients (*n* = 57)
Hard palate_RESISTANCE (Mean ± SD [N])	0.044	75.7 ± 13.8 (*n* = 60)	80.6 ± 16.7 (*n* = 57)
Hard palate_RESISTANCE (Median [min–max])		74.2 (46.0–108.2)	82.4 (16.2–110.8)
Hard palate_REACTANCE (Mean ± SD [N])	0.020	14.5 ± 5.5 (*n* =60)	17.0 ± 6.3 (*n* = 57)
Hard palate_REACTANCE (Median [min–max])		13.9 (4.8–36.8)	15.9 (6.1–35.2)
Hard palate_PHASE ANGLE (Mean ± SD [N])	0.054	10.7 ± 3.6 (*n* = 60)	12.7 ± 7.7 (*n* = 57)
Hard palate_PHASE ANGLE (Median [min–max])		9.4 (5.3–23.9)	11.2 (6.3–62.0)
Hard palate_TEMPERATURE (Mean ± SD [N])	<0.0001	25.1 ± 1.0 (*n* = 60)	23.9 ± 1.4 (*n* = 57)
Hard palate_TEMPERATURE (Median [min–max])		25.3 (22.5–27.7)	23.6 (21.2–26.8)
Adherent gingiva_RESISTANCE (Mean ± SD [N])	<0.0001	59.0 ± 13.8 (*n* = 60)	78.6 ± 30.5 (*n* = 57)
Adherent gingiva_RESISTANCE (Median [min–max])		57.2 (32.2–116.5)	72.9 (31.4–188.6)
Adherent gingiva_REACTANCE (Mean ± SD [N])	<0.0001	13.6 ± 5.7 (*n* = 60)	19.6 ± 11.8 (*n* = 57)
Adherent gingiva_REACTANCE (Median [min–max])		12.3 (5.9–38.5)	16.2 (5.4–66.6)
Adherent gingiva_PHASE ANGLE (Mean ± SD [N])	0.12	12.7 ± 2.9 (*n* = 60)	13.5 ± 3.0 (*n* = 57)
Adherent gingiva_PHASE ANGLE (Median [min–max])		12.5 (5.8–21.6)	13.5 (5.8–19.7)
Adherent gingiva_TEMPERATURE (Mean ± SD [*n*])	<0.0001	25.1 ± 1.0 (*n* = 60)	24.0 ± 1.5 (*n* = 57)
Adherent gingiva_TEMPERATURE (Median [min–max])		25.3 (22.6–27.7)	23.7 (21.1–26.9)
Left cheek_RESISTANCE (Mean ± SD [N])	0.12	65.9 ± 10.7 (*n* = 60)	62.5 ± 9.4 (*n* = 57)
Left cheek_RESISTANCE (Median [min–max])		64.8 (47.1–96.5)	62.5 (45.1–89.8)
Left cheek_REACTANCE (Mean ± SD [N])	0.63	13.8 ± 2.2 (*n* = 60)	13.8 ± 2.9 (*n* = 57)
Left cheek _REACTANCE (Median [min–max])		14.0 (8.9–19.3)	13.4 (7.2–20.5)
Left cheek_PHASE ANGLE (Mean ± SD [N])	0.11	11.9 ± 1.3 (*n* = 60)	12.5 ± 1.9 (*n* = 57)
Left cheek_PHASE ANGLE (Median [min–max])		12.0 (9.1–15.8)	12.2 (9.1–16.4)
Left cheek_TEMPERATURE (Mean ± SD [N])	<0.0001	24.9 ± 0.9 (*n* = 60)	23.6 ± 1.6 (*n =* 57)
Left cheek_TEMPERATURE (Median [min–max])		25.0 (22.8–27.2)	23.4 (20.4–26.8)
Right cheek_RESISTANCE (Mean ± SD [N])	0.10	66.6 ± 8.9 (*n =* 60)	64.0 ± 9.9 (*n =* 57)
Right cheek_RESISTANCE (Median [min–max])		66.7 (49.4–85.9)	63.6 (45.3–92.6)
Right cheek_REACTANCE (Mean ± SD [N])	0.091	14.6 ± 2.4 (*n =* 60)	13.9 ± 2.5 (*n =* 57)
Right cheek_REACTANCE (Median [min–max])		14.6 (9.0–21.0)	13.5 (8.5–21.9)
Right cheek_PHASE ANGLE (Mean ± SD [N])	0.93	12.3 ± 2.0 (*n =* 60)	12.2 ± 1.5 (*n =* 57)
Right cheek_PHASE ANGLE (Median [min–max])		12.1 (8.0–20.9)	12.1 (9.9–17.7)
Right cheek_TEMPERATURE (Mean ± SD [N])	<0.0001	24.9 ± 0.9 (*n =* 60)	23.5 ± 1.6 (*n =* 57)
Right cheek_TEMPERATURE (Median [min–max])		25.0 (22.7–27.3)	23.4 (20.3–26.7)
Tongue_RESISTANCE (Mean ± SD [N])	0.72	57.3 ± 8.3 (*n =* 60)	58.3 ± 11.1 (*n =* 57)
Tongue_RESISTANCE (Median [min–max])		56.8 (35.9–76.7)	55.4 (46.0–110.3)
Tongue_REACTANCE (Mean ± SD [N])	0.35	15.0 ± 2.6 (*n =* 60)	14.7 ± 2.7 (*n =* 57)
Tongue_REACTANCE (Median [min–max])		14.6 (7.8–20.1)	14.4 (9.8–24.9)
Tongue_PHASE ANGLE (Mean ± SD [N])	0.19	14.5 ± 1.2 (*n =* 60)	14.3 ± 1.5 (*n =* 57)
Tongue_PHASE ANGLE (Median [min–max])		14.6 (11.2–17.4)	14.4 (11.3–19.1)
Tongue_TEMPERATURE (Mean ± SD [N])	<0.0001	25.0 ± 0.9 (*n =* 60)	23.8 ± 1.6 (*n =* 57)
Tongue_TEMPERATURE (Median [min–max])		25.2 (22.9–27.6)	23.5 (20.8–26.8)

**Table 2 dentistry-10-00137-t002:** Comparison between measurements in different types of lesion in OLP patients (left cheek), classified as: no lesions = 0; striations = 1; erythema or erosion = 2. Electrical values are in Ohm. 1= Wilcoxon’s rank-sum test.

Variable	*p* Value	0	1	2
		(*n =* 21)	(*n =* 25)	(*n =* 10)
Left cheek_RESISTANCE (Mean ± SD [N])	0.37	63.2 ± 10.4 (*n =* 21)	63.9 ± 8.8 (*n =* 25)	58.3 ± 8.8 (*n =* 10)
Left cheek_RESISTANCE (Median [min–max])		63.0 (47.9–89.8)	62.5 (49.1–82.2)	61.5 (45.1–70.8)
Left cheek_REACTANCE (Mean ± SD [N])	0.10	13.8 ± 2.6 (*n =* 21)	14.6 ± 2.9 (*n =* 25)	12.2 ± 3.1 (*n =* 10)
Left cheek_REACTANCE (Median [min–max])		12.7 (10.6–20.5)	14.0 (10.3–20.5)	11.8 (7.2–18.5)
Left cheek_PHASE ANGLE (Mean ± SD [N])	0.29	12.3 ± 1.6 (*n =* 21)	12.9 ± 2.0 (*n =* 25)	11.9 ± 2.3 (*n =* 10)
Left cheek_PHASE ANGLE (Median [min–max])		12.2 (9.3–14.5)	12.6 (9.2–16.3)	11.4 (9.1–16.4)
Left cheek_TEMPERATURE (Mean ± SD [N])	0.18	23.5 ± 1.5 (*n =* 21)	23.5 ± 1.6 (*n =* 25)	24.4 ± 1.5 (*n =* 10)
Left cheek_TEMPERATURE (Median [min–max])		23.3 (21.0–26.8)	23.4 (20.4–26.4)	24.5 (21.5–26.8)

**Table 3 dentistry-10-00137-t003:** Comparison between measurements in different types of lesion in OLP patients (right cheek), classified as: no lesions = 0; striations = 1; erythema or erosion = 2. Electrical values are in Ohm. 1= Wilcoxon’s rank-sum test.

Variable	*p* Value	0	1	2
		(*n =* 19)	(*n =* 28)	(*n =* 9)
Right cheek_RESISTANCE (Mean ± SD [N])	0.90	64.2 ± 8.9 (*n =* 19)	63.8 ± 9.7 (*n =* 28)	64.2 ± 13.8 (*n =* 9)
Right cheek_RESISTANCE (Median [min–max])		62.0 (52.5–83.1)	62.7 (46.6–92.6)	68.3 (45.3–81.6)
Right cheek_REACTANCE (Mean ± SD [N])	0.58	14.4 ± 2.5 (*n =* 19)	14.0 ± 2.3 (*n =* 28)	12.6 ± 2.8 (*n =* 9)
Right cheek_REACTANCE (Median [min–max])		13.3 (11.8–21.9)	13.5 (9.8–19.8)	13.3 (8.5–16.7)
Right cheek_PHASE ANGLE (Mean ± SD [N])	0.0099	12.6 ± 1.2 (*n =* 19)	12.4 ± 1.7 (*n =* 28)	11.1 ± 0.6 (*n =* 9)
Right cheek_PHASE ANGLE (Median [min–max])		12.4 (10.9–15.1)	12.3 (9.9–17.7)	10.9 (10.5–12.1)
Right cheek_TEMPERATURE (Mean ± SD [N])	0.24	23.5 ± 1.7 (*n =* 19)	23.3 ± 1.6 (*n =* 28)	24.3 ± 1.2 (*n =* 9)
Right cheek_TEMPERATURE (Median [min–max])		23.1 (20.3–26.7)	23.4 (20.8–26.4)	24.0 (23.0–26.5)

**Table 4 dentistry-10-00137-t004:** Comparison between types of lesions in the right cheek using the Dwass–Steel–Critchlow–Fligner multiple comparison procedure.

	Wilcoxon Z	DSCF Value	Pr > DSCF
No lesions vs. Striations	0.6509	0.9205	0.7918
No lesions vs. Erythema or Erosion	3.1275	4.423	0.005
Striations vs. Erythema or Erosion	2.4279	3.4336	0.0403

**Table 5 dentistry-10-00137-t005:** Comparison between measurements in different types of lesion in OLP patients (tongue), classified as: no lesions = 0; striations = 1; erythema or erosion = 2. Electrical values are in Ohm. 1 = Wilcoxon’s rank-sum test.

Variable	*p* Value	0	1
		(*n =* 43)	(*n =* 12)
Tongue_RESISTANCE (Mean ± SD [N])	0.30	59.0 ± 12.3 (*n =* 43)	55.1 ± 5.3 (*n =* 12)
Tongue_RESISTANCE (Median [min–max])		56.7 (46.0–110.3)	54.0 (48.0–67.0)
Tongue_REACTANCE (Mean ± SD [N])	0.91	14.7 ± 2.8 (*n =* 43)	14.4 ± 1.6 (*n =* 12)
Tongue_REACTANCE (Median [min–max])		14.4 (9.8–24.9)	14.2 (11.6–17.3)
Tongue_PHASE ANGLE (Mean ± SD [N])	0.39	14.1 ± 1.6 (*n =* 43)	14.5 ± 0.7 (*n =* 12)
Tongue_PHASE ANGLE (Median [min–max])		14.1 (11.3–19.1)	14.6 (13.2–15.5)
Tongue_TEMPERATURE (Mean ± SD [N])	0.28	23.9 ± 1.6 (*n =* 43)	23.4 ± 1.5 (*n =* 12)
Tongue_TEMPERATURE (Median [min–max])		23.6 (21.3–26.8)	23.2 (21.1–25.6)

## Data Availability

The data presented in this study are available on request from the corresponding author.

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
