# Peer review of "Bioelectrical Impedance Analysis of Oral Cavity Mucosa in Patients with Lichen Planus and Healthy Controls"

_dentistry, 2022, doi:10.3390/dj10070137_

Round 1
Reviewer 1 Report
The article is written correctly. The proposed diagnostic technique is innovative even though it is already applied in other branches of Medicine. The technique is minimally invasive. The results is encouraging. The technique could be applied to other mucosal lesions.
Author Response
Comments and Suggestions for Authors
The article is written correctly. The proposed diagnostic technique is innovative even though it is already applied in other branches of Medicine. The technique is minimally invasive. The results is encouraging. The technique could be applied to other mucosal lesions.
Reply: Thank You so much!
Text changes have been highlighted in yellow
Reviewer 2 Report
Thank you for giving me the opportunity to review the paper of Bacci et al. the major aim of this paper is to perform Bioelectrical Impedance Analyses to assess the difference between oral lichen planus and normal oral mucosa.
The original idea of the paper is nice. However, the work was not presented in a good way and in my opinion, the current format is far away from being considered solid enough for publication.
The ideas are difficult to follow and readers have to repeat sections several times to understand the idea. The used English language needs comprehensive improvement.
I could not comment on each section because doing that means that I need to write a new manuscript. However, the following are some comments that may help the authors improve their manuscript.
Abstract
· Line 12: please add more details about OLP, only one or two sentences would be enough. for example, OLP is a relatively chronic inflammatory condition with an unknown etiology, OR without diagnostic features. You can add whatever you want as long as it is related to the topic of the paper
· Line 12: here the authors immediately started talking about the BIA without providing a suitable introduction. I would suggest providing a sentence about the tissue composition or imbalance in OLP before talking about BIA. the current abstract lacks the rationale behind conducting this study. So as a reader, I want to know from the beginning why this work is important and what is the benefit of conducting such a project.
· Line 19: what was this specific device, oral brush, biopsy, light ??
· Line 21: what does PA refer to?
· Line 22: please provide the p values
· There are no actual mentioned results to support the conclusion.
Introduction
· The introduction in general is very wordy. In my opinion, there is no need to talk about the clinical presentations of OLP in detail. Line 45 onwards talk about technical details that are not supposed to be in an introduction.
· Again, there is no rationale to support the study. No literature review about the benefit of using this technique in diagnosis, either in dentistry or other medical fields.
· Line 31: please add a citation to support this part
· Line 33: there is no single site OLP, it is either typical bilateral/symmetrical OLP or oral lichenoid lesion
· Line 35: reference 4: please try to use the most recent classification in this area of Cheng et al in 2016 https://pubmed.ncbi.nlm.nih.gov/27401683/
the current reference is very old, 1968
· Line 38: please cite papers to support the malignant transformation rate
· Line 114: we cannot say that erosive OLP mimics tumours, OLP is OLP and tumour is tumour and between them, there are thousands of biological and molecular changes.
Methods
· Line 119: in the abstract is written 57 OLP cases, why there is a difference in the numbers of patients.
· Line 130: the authors mentioned that the control group matched with the age and gender of the OLP group, however, in the result section I cannot see that both groups are matched.
· Line 135: what do the authors mean by “In the graphs and tables, any lesions are identified by means of numerical values”?
· Line 142: I don’t think that the oral cavity temperature is strongly influenced by room temperature !!!
Results
· Line 171: these are new numbers.
· Line 184 to 186: these details are very wordy and not necessary
· Line 204 to 208: these details are already mentioned above, there is no need for repeating.
· Line 188: I would suggest that the authors mentioned the detected number of lesions rather than the number of regions without lesions. For example, what is the benefit for me as a reader to know that 53 patients had no lesions in the hard palate?? I want to know how many lesions were on hard palate
· Line 211: what are these p values? are they for the attached gingiva and palate? if yes, then we should have only 2 values. moreover, the last p-value (p=0.054) is not significant
· Tables: it looks like Wilcoxon test was used for all analyses in all tables, then there is no need to add "1" after each p-value
· Table 4: this should be dot "." rather than comma ","
Author Response
(x) Extensive editing of English language and style required
Reply: The paper was corrected by a native speaker and double checked
Comments and Suggestions for Authors
Thank you for giving me the opportunity to review the paper of Bacci et al. the major aim of this paper is to perform Bioelectrical Impedance Analyses to assess the difference between oral lichen planus and normal oral mucosa.
Reply: Thank You!
The original idea of the paper is nice. However, the work was not presented in a good way and in my opinion, the current format is far away from being considered solid enough for publication.
Reply: this paper reports an experimental study. Anyway a similar paper is still present in the literature even (but this is my personal opinion) its methodology is not so good… (Tatullo, M., Marrelli, M., Amantea, M., Paduano, F., Santacroce, L., Gentile, S. and Scacco, S., 2015. Bioimpedance detection of oral lichen planus used as preneoplastic model. Journal of Cancer, 6(10), p.976.)
The ideas are difficult to follow and readers have to repeat sections several times to understand the idea. The used English language needs comprehensive improvement.
Reply: I apologize for my English but for this reason the paper was corrected by a native speaker and double checked
I could not comment on each section because doing that means that I need to write a new manuscript. However, the following are some comments that may help the authors improve their manuscript.
Abstract
- Line 12: please add more details about OLP, only one or two sentences would be enough. for example, OLP is a relatively chronic inflammatory condition with an unknown etiology, OR without diagnostic features. You can add whatever you want as long as it is related to the topic of the paper
Reply: So in your opinion, we wrote too much in the introduction while too little in the abstract?
Giving a summary and univocal definition of lichen in a few words is complex and many other papers summarize with "Oral lichen planus (OLP) is an inflammatory disease" in the abstract and then specify in the text.
- Line 12: here the authors immediately started talking about the BIA without providing a suitable introduction. I would suggest providing a sentence about the tissue composition or imbalance in OLP before talking about BIA. the current abstract lacks the rationale behind conducting this study. So as a reader, I want to know from the beginning why this work is important and what is the benefit of conducting such a project.
Reply: Page 2 contains an extensive description of the BIA
- Line 19: what was this specific device, oral brush, biopsy, light ??
Reply: the medical device is BIA
- Line 21: what does PA refer to?
Reply: phase angle, I apologize
- Line 22: please provide the p values
Reply: inserted
- There are no actual mentioned results to support the conclusion.
Reply: Please see the tables.
Introduction
- The introduction in general is very wordy. In my opinion, there is no need to talk about the clinical presentations of OLP in detail. Line 45 onwards talk about technical details that are not supposed to be in an introduction.
Reply: So in your opinion, we wrote too much in the introduction while too little in the abstract?
- Again, there is no rationale to support the study. No literature review about the benefit of using this technique in diagnosis, either in dentistry or other medical fields.
Reply: Of course, this is an experimental study (with positive outcomes anyway)
- Line 31: please add a citation to support this part
- Line 33: there is no single site OLP, it is either typical bilateral/symmetrical OLP or oral lichenoid lesion
Reply: clinical aspect may be different by side
- Line 35: reference 4: please try to use the most recent classification in this area of Cheng et al in 2016 https://pubmed.ncbi.nlm.nih.gov/27401683/
Reply: Thank You for this suggest!! I’ve inserted it even in diagnostic criteria…
- the current reference is very old, 1968
Reply: Thank for this suggest, too… It has been deleted.
- Line 38: please cite papers to support the malignant transformation rate
Reply: Done
- Line 114: we cannot say that erosive OLP mimics tumours, OLP is OLP and tumour is tumour and between them, there are thousands of biological and molecular changes.
Reply: Modified as suggested.
Methods
- Line 119: in the abstract is written 57 OLP cases, why there is a difference in the numbers of patients.
Reply: 63 were enrolled, then 6 excluded. 57 took part in the trial.
- Line 130: the authors mentioned that the control group matched with the age and gender of the OLP group, however, in the result section I cannot see that both groups are matched.
Reply: “mean 60.3 years old (minimum 23, maximum 86 years), and the control group was a mean 63.1 years old (minimum 23, maximum 89)” (line 180)
- Line 135: what do the authors mean by “In the graphs and tables, any lesions are identified by means of numerical values”?
Reply: Modified
- Line 142: I don’t think that the oral cavity temperature is strongly influenced by room temperature !!!
Reply: Maybe, but at my best knowledge, no literature is available on this topic.
Results
- Line 171: these are new numbers.
Reply: Of course!! Is the Results section…
- Line 184 to 186: these details are very wordy and not necessary
Reply: I am willing to delete these sentences but the Results section would be less clear.
- Line 204 to 208: these details are already mentioned above, there is no need for repeating.
Reply: Modified as suggested.
- Line 188: I would suggest that the authors mentioned the detected number of lesions rather than the number of regions without lesions. For example, what is the benefit for me as a reader to know that 53 patients had no lesions in the hard palate?? I want to know how many lesions were on hard palate
Reply: This data organization were established with the statistic expert.
- Line 211: what are these p values? are they for the attached gingiva and palate? if yes, then we should have only 2 values. moreover, the last p-value (p=0.054) is not significant
Reply: “The level of significance was set at 5%” (the same p=0.05) (line 168)
- Tables: it looks like Wilcoxon test was used for all analyses in all tables, then there is no need to add "1" after each p-value
Reply: Modified as suggested
- Table 4: this should be dot "." rather than comma ","
Reply: Modified as suggested
Text changes have been highlighted in yellow
Reviewer 3 Report
· That paper is relevant and interesting
· The topic is original and the paper is well written . The text is clear and easy to read
· The introduction is complete and focuses on the problem of the study
· The results and discussions of the manuscript are adequate
· the conclusions are consistent with the evidence and arguments presented .
· The literature collected in this study is adequate
I should mention the experimentation ethics committee that does not appear in the text.
This is a clinical study so you must follow the consort guide . Serious methodological problems
It does not have a clinical record ( gove trial )
Criteria for lichen planus should be defined
Who carried out the study; skilled person; It is not clear if they are in reticular forms or erosive forms
sample size ¿?
Author Response
Comments and Suggestions for Authors
- That paper is relevant and interesting
- The topic is original and the paper is well written . The text is clear and easy to read
- The introduction is complete and focuses on the problem of the study
- The results and discussions of the manuscript are adequate
- the conclusions are consistent with the evidence and arguments presented .
- The literature collected in this study is adequate
Reply: Thank You so much!
I should mention the experimentation ethics committee that does not appear in the text.
Reply: At page 13 You can find “Compliance with ethical standards”.
This is a clinical study so you must follow the consort guide . Serious methodological problems
It does not have a clinical record ( gove trial )
Reply: No, this study did not but received clearance from the local ethics committee.
Criteria for lichen planus should be defined
Reply: Inserted.
Who carried out the study; skilled person; It is not clear if they are in reticular forms or erosive forms
Reply: Skilled person. Both (reticular and erosive) (page 4)
sample size ¿?
Reply: The sample size is not low (127 subjects) but in any case statistical significance is achieved.
And, as stated: “numerosity of the sample required was established a priori.”
Text changes have been highlighted in yellow
Round 2
Reviewer 2 Report
1. There are still some English errors, please check the English language of the whole paper carefully.
2. When I asked the authors what PA in the abstract means the authors answered “phase angle”, and when I asked the authors to include more representable results in the abstract the authors responded, “Please see the tables”. The responses are too brief and not professional. Some major issues have not been addressed well.
3. This paper contains flaws contradicting the global consensus regarding the diagnosis of OLP (being bilateral and/or symmetrical).
Author Response
- There are still some English errors, please check the English language of the whole paper carefully.
Reply: The paper has been reviewed and double checked by a native.
- When I asked the authors what PA in the abstract means the authors answered “phase angle”
Reply: …and the text has been modified…
and when I asked the authors to include more representable results in the abstract the authors responded, “Please see the tables”. The responses are too brief and not professional. Some major issues have not been addressed well.
Reply: I apologize: I was wrong; I didn't think it was an annotation made to the abstract but to the text.
Modified as suggested.
- This paper contains flaws contradicting the global consensus regarding the diagnosis of OLP (being bilateral and/or symmetrical)
Reply: Yes, the clinical appearance of the OPL may change over time and the lesions become more evident at different times.

Reviewer 3 Report
The changes have been implemented satisfactorily
Author Response
The changes have been implemented satisfactorily
Reply: Thank You so much!
